# Circulating miRNome detection analysis reveals 537 miRNAS in plasma, 625 in extracellular vesicles and a discriminant plasma signature of 19 miRNAs in children with retinoblastoma from which 14 are also detected in corresponding primary tumors

**Blanca Elena Castro-Magdonel[1], Manuela Orjuela[2], Diana E. Alvarez-Suarez[1,3], Javier Camacho[3], Lourdes Cabrera-Muñoz[4], Stanislaw Sadowinski-Pine[4], Aurora Medina-Sanson[5], Citlali Lara-Molina[6], Daphne García-Vega[7], Yolanda Vázquez[7], Noé Durán-Figueroa[8], María de Jesús Orozco-Romero[8], Adriana Hernández-Ángeles[1], M. Verónica Ponce-Castañeda[1] ***

1 Medical Research Unit in Infectious Diseases, Hospital de Pediatría, CMN SXXI, Instituto Mexicano del Seguro Social, Mexico City, Mexico, 2 Epidemiology Department, Columbia University, New York, New York, United States of America, 3 Pharmacology Department, CINVESTAV, Mexico City, Mexico, 4 Pathology Department, Hospital Infantil de Mexico Federico Gómez, Secretaría de Salud, Mexico City, Mexico, 5 Oncology Department, Hospital Infantil de México Federico Gómez, Secretaría de Salud, Mexico City, Mexico, 6 Ophtalmology Department, Hospital Infantil de Mexico Federico Gómez, Secretaría de Salud, Mexico City, Mexico, 7 Ophtalmology Department, Hospital de Pediatría Silvestre Frenk, CMN SXXI, IMSS, Mexico City, Mexico, 8 Unidad Profesional Interdisciplinaria de Biotecnología, Instituto Politécnico Nacional, Mexico City, Mexico

* vponce@ifc.unam.mx

## Abstract

miRNAs regulate post-transcriptional gene expression in metazoans, and thus are involved in many fundamental cellular biological processes. Extracellular miRNAs are also found in most human biofluids including plasma. These circulating miRNAs constitute a long distance inter cellular communication system and are potentially useful biomarkers. High throughput technologies like microarrays are able to scan a complete miRNome providing useful detection scores that are underexplored. We proposed to answer how many and which miRNAs are detectable in plasma or extracellular vesicles as these questions have not yet been answered. We set out to address this knowledge gap by analyzing the mirRNome in plasma and corresponding extracellular vesicles (EVs) from 12 children affected by retinoblastoma (Rb) a childhood intraocular malignant tumor, as well as from 12 healthy similarly aged controls. We calculated an average of 537 detectable miRNAs in plasma and 625 in EVs. The most miRNA enriched compartment were EVs from Rb cases with an average of 656 detectable elements. Using hierarchical clustering with the detection scores, we generated broad detection mirnome maps and identified a plasma signature of 19 miRNAs present in all Rb cases that is able to discriminate cases from controls. An additional 9 miRNAs were detected in all the samples; within this group, miRNA-5787 and miRNA-6732-5p were

**Data Availability Statement:** The data underlying the results presented in the study are available from NCBI at https://www.ncbi.nlm.nih.gov/geo/query/acc.cgi?acc=GSE141208.

**Funding:** This work has been funded by CONACYT FOSISS-I010/270/13 to NVDF https://www.conacyt.gob.mx/ Also by the National Institutes of Health, USA, grants CA167833, CA192662, CA98180 to MO https://www.nih.gov/.

**Competing interests:** The authors have declared that no competing interests exist.

highly abundant and displayed very low variance across all the samples, suggesting both are good candidates to serve as plasma references or normalizers. Further exploration considering participant's sex, allowed discovering 5 miRNAs which corresponded only to females and 4 miRNAs corresponding only to males. Target and pathway analysis of these miRNAs revealed hormonal function including estrogen, thyroid signaling pathways and testosterone biosynthesis. This approach allows a comprehensive unbiased survey of a circulating miRNome landscape, creating the possibility to define normality in mirnomic profiles, and to locate where in these miRNome profiles promising and potentially useful circulating miRNA signatures can be found.

## Introduction

miRNAs are an important class of small RNAs discovered in *C. elegans* in 1993. They have been detected in plants, animals and some viruses. Inside cells, miRNAs are mostly down regulators of gene expression, interfering with the process of mRNA translation into proteins [1] [2][3]. In humans, approximately 2600 mature miRNAs have been reported in the curated miRNAs registry, the miRBase [4]. Many of the human miRNAs have only recently been identified using miRNAseq. Consequently, many of these miRNA's have not yet had experimentally supported biological functions attributed to them [5]. These small molecules have also been found circulating in human plasma, serum and in other body fluids including saliva, milk, urine, and semen. In plasma, miRNAs associate with Argonaut proteins and can also be detected inside extracellular vesicles or exosomes [6][7].

In plasma, exosomes are part of extracellular vesicles (EVs) which include particles of diverse sizes and origins that are not fully characterized. There is no standard definition for what exosomes are. In practical terms, these are mixed populations of membranous nanosized particles about 50–130 nm in diameter [8][9][10]. They contain biochemically diverse molecules including miRNAs, long non coding RNA, DNA and proteins from the cell from which they originate and are actively released. Nonetheless, EVs are able to change the transcriptional context of distant target cells by fusing their membrane to that of the target cell and releasing their cargo within the cell [11]. EVs and Argonaut proteins protect miRNAs from RNAse present in plasma, rendering miRNAs very stable in blood [12][13]. The functional convergence of EVs and miRNAs constitutes a long distance cell communication system that has been recognized as such during the last decade [14]. Currently, there is active ongoing research into the biochemical and functional characterization of EV contents and their membrane proteins. EVs have also been implicated in the tissue specific tropism of metastatic processes in breast cancer [10][15][16]. Part of the interest in EVs stems from their potential as a rich source of biomarkers using minimally invasive techniques. Although there is no direct path for developing clinically useful biomarkers, some recently reported attributes of miRNA profiles in plasma or serum make them good candidates for use as biomarkers. Examples of these include profiles in cancer patients that can differentiate different types of tumors, or tumor profile correlations with tumor progression [17][18][12].

### Microarrays for analyzing the miRNome

High-throughput expression microarray technology can now scan complete miRNnomes or transcriptomes in a single hybridization experiment. The original technology used a double

channel platform and yielded relative measurements. However, more mature technology with one-channel platforms include calibrators, which yield absolute measurements. These calibrators allowed the development of robust methods for detection calls [19], which are basically ignored by most analysis. Broad maps of what is detected and what is not can be generated with the detection score using clustering tools thus facilitating data interpretation [20]. This approach allows answering relevant questions, such as ´how many´ and ´which´ miRNAs can be detected in plasma or EVs? Similarly, one can now posit: ´Are detectable miRNAs shared by all samples, phenotypes or experimental conditions? ´

Here, using the detection score that microarrays analysis provide, we report results obtained with panoramic and simplified mirnomic graphs from plasma and corresponding EVs, allowing us to determine miRNA contents in these two blood compartments. Using unsupervised hierarchical clustering with Venn diagrams [21], and contrasting them with differential analysis, we discovered miRNA clusters shared by all the children with retinoblastoma [Rb] cases, clusters shared by all healthy controls, one small cluster shared by all samples and some miRNAs related to sex. We further discuss the relationship of these miRNAs with the corresponding tumor miRNA profiles that we had previously reported, and propose that those miRNAs consistently detected in all samples can serve as novel circulating normalizers.

## Methods

This research was approved by Comisión Nacional de Investigación Científica and Comisión de Bioética IMSS. Also by the Comité Local de Investigación y Bioética, Hospital de Pediatría Silvestre Frenk Freund, CMN SXXI, IMSS.

### Patients and samples

We examined samples from 12 children with retinoblastoma and 12 children without retinoblastoma and all children were of Hispanic ethnicity. Inclusion criteria: children newly diagnosed with retinoblastoma at an age of less than or equal to 6 years since most affected patients are in this age group. Without a known genetic syndrome or prior diagnosis of cancer, who were diagnosed or were receiving treatment at the two participating treatment centers, and did not have a preexisting or known family history of retinoblastoma. One child later found to have familial retinoblastoma was included in the study. Controls were healthy children with the same inclusion criteria as cases, who were recruited as friend controls as previously described [22]. Exclusion criteria: age greater than 6 years, known diagnosis of a genetic syndrome or prior diagnosis of cancer.

All samples were collected after obtaining written informed parental consent to participate in a larger IRB approved case-control study. Because of the age of children, assent was not required. The children were patients diagnosed and receiving treatment at Hospital de Pediatría Silvestre Frenk at Centro Médico Nacional Siglo XXI, from Instituto Mexicano del Seguro Social (IMSS), and Hospital Infantil de México Federico Gómez, from Secretaría de Salud in Mexico City during the years 2004 to 2012 as previously described [22][23].

### Plasma samples

All blood samples were obtained prior to the child's receiving any chemotherapy or radiation, and were drawn into Becton Dickinson CPT tubes (Becton Dickinson, Franklin Lakes, NJ USA) with citrate as anticoagulant and stored at 4˚C until separation by centrifugation. Samples were fractionated within 24 hours of collection. Plasma aliquots of 1 ml were stored at -80˚ C until processed for isolation of EVs and RNA extraction.

### Extracellular vesicles (EVs)

EVs were isolated from 0.5 ml of each plasma sample with an initial centrifugation at 2000 x g to remove any insoluble material. Supernatant was then diluted with PBS 2:1 and filtered with a syringe driven filter unit 0.22-micron pore, (Millex, Millipore SLGV033RS). Each resulting sample was mixed with 5:1 Exosome Isolation Reagent (Invitrogen 4478359), incubated on ice for 10 minutes and then centrifuged at 10,000 x g for 30 minutes. Precipitated material was resuspended in 100 microliters (µl) PBS and stored at -20˚C until processed for RNA isolation. We used electron microscopy to visualize EVs [24].

### RNA isolation and labeling for microarrays

Total RNA was isolated from 0.5 ml of each plasma sample, and from 100 µl of purified EVs using TRIzol reagent (Invitrogen 15596026), solubilized in nuclease free water and quantified using a Nanodrop. RNA 150 ng was poly-A tailed and biotin incorporated into tails with Flash Tag biotin for miRNA microarrays (Affymetrix 901911), including an Enzyme Linked Oligo-sorbent Assay (ELOSA) to confirm biotin labeling according to manufacturer instructions and as previously described [20].

### miRNA microarrays

Hybridization mix was added to labeled samples containing 2X hybridization mix, 27% form-amide, DMSO and 20X hybridization control probes in nuclease free water. Each sample was introduced into a GeneChip miRNA 4 array (Affymetrix 902411), hybridized at 48˚C for 18 h. Each sample was washed in the GenChip Fluidics Station 450 following FS450-0002 protocol, and array fluorescence was measured by the Gene Chip Scanner 3000 7 G

### Data analysis

Demographic characteristics of the sample of 12 children with Rb and the 12 healthy controls were compared with Mann Whitney and Chi Square tests using SPSS v26. Raw data from .CEL files with fluorescence signals were preprocessed in Affymetrix Expression Console software (v 1.4.1.46; Affymetrix UK Ltd). Preprocessing included normalization, quality control and detection call. We used Robust Multichip Analysis (RMA) to adjust the signal to noise ratio and to generate CHP files for further analysis. For each probeset or miRNA in the array we obtained two types of data from .CHP files: 1) detection score, categorizing each miRNA as 'present' or 'absent' and 2) detection levels log2 intensity signal data. Algorithms used for detection call metrics were developed and published by the supplier using the Latin Square experimental design; on which artificial transcripts at known concentrations were spiked in a complex background [19]. Analysis from this design generates a detection score with an associated $p$ value that reflects the confidence of the detected ('present') or not detected ('absent') score, and is very useful when the detection level of a miRNA is close to the local background.

### Bioinformatic analysis

Detection scores were analyzed and grouped with unsupervised hierarchical clustering (HC) in Multi Experiment Viewer (MEV) from TM4 Microarray Software Suite [25]. Differential analysis was performed in R (R core team, 2019) with Bioconductor software package Limma [26] incorporating correction for multiple testing. miRNAs with p-values < 0.05 were grouped by hierarchical clustering and plotted with the R gplots [27]. To search for mRNA targets and signaling pathways involved with specific miRNAs we used DIANA-Tools-microT-CDS,

miRPathv3, Target-Scan and Tarbase v.7 (at http://diana.imis.athena-innovation.gr/Diana-Tools/index.php?r_tarbase/index) [28].

## Results

Demographic and clinical characteristics of the participants are shown in **Table 1**. Cases and controls were comparable in their distribution of age at time of sample collection as well as in their sex distribution.

### Mirnomic landscape identify common miRNAs in plasma, EVs and primary tumors from Rb patients

To test if plasma and EVs from Rb cases share miRNA elements with the primary tumors, we generated the corresponding plasma and EV datasets and compared these with a publicly available mirnomic dataset of the corresponding tumor tissues that we had previously reported (GEO GSE 84747 and Array Express E-MTAB-4977) [20]. With unsupervised hierarchical clustering analysis (clustering both samples and miRNAs), we generated a miRNA detection two-color heatmap of these samples shown in **Fig 1**.

This map shows a panoramic mirnomic view of detected and not detected miRNAs in all samples derived from Rb cases. Near the top there is a small cluster of 18 miRNAs shared by all the samples, indicating these miRNAs detected in the primary tumor, are also detected in circulation (i.e., in plasma and in EV) suggesting the existence of a circulating tumor signature. In this detection map, all retinoblastoma tumor tissues group together, while most plasmas group in the center and most EVs group to the right, although their distribution does not segregate completely. This map shows a higher density of detected miRs in tumors as expected, and there appears to be a higher density of miRNAs in EVs than in plasma.

After obtaining these results, we decided to analyze each circulating compartment separately in order to characterize similarities and differences between children with Rb and healthy controls.

Table 1. Select demographic characteristics of the Rb patients and healthy controls whose samples were analyzed.

| Rb patient | Age (months) | Sex | Laterality^ | Healthy Control | Age (months) | Sex |
|---|---|---|---|---|---|---|
| P1 | 26 | F | B | C1 | 9 | F |
| P2 | 32 | M | B | C2 | 52 | M |
| P3 | 29 | M | B | C3 | 41 | M |
| P4 | 60 | M | U | C4 | 28 | F |
| P5 | 37 | F | U | C5 | 30 | F |
| P6 | 52 | F | U | C6 | 71 | M |
| P7 | 19 | M | U | C7 | 15 | M |
| P8 | 9 | M | B | C8 | 22 | M |
| P9 | 12 | M | B | C9 | 4 | M |
| P10 | 36 | M | U | C10 | 17 | F |
| P11 | 33 | M | U | C11 | 57 | F |
| P12 | 36 | F | U | C12 | 67 | M |
| Median (min, max) | 31.9 (9, 61.9) | | | | 29.3 (9, 71.5) | |

^ B is bilateral; U is unilateral.

Cases and controls did not differ in age (Mann-Whitney test based on ages in months at the time of sample collection) or in distribution of sex (by Chi Square test).

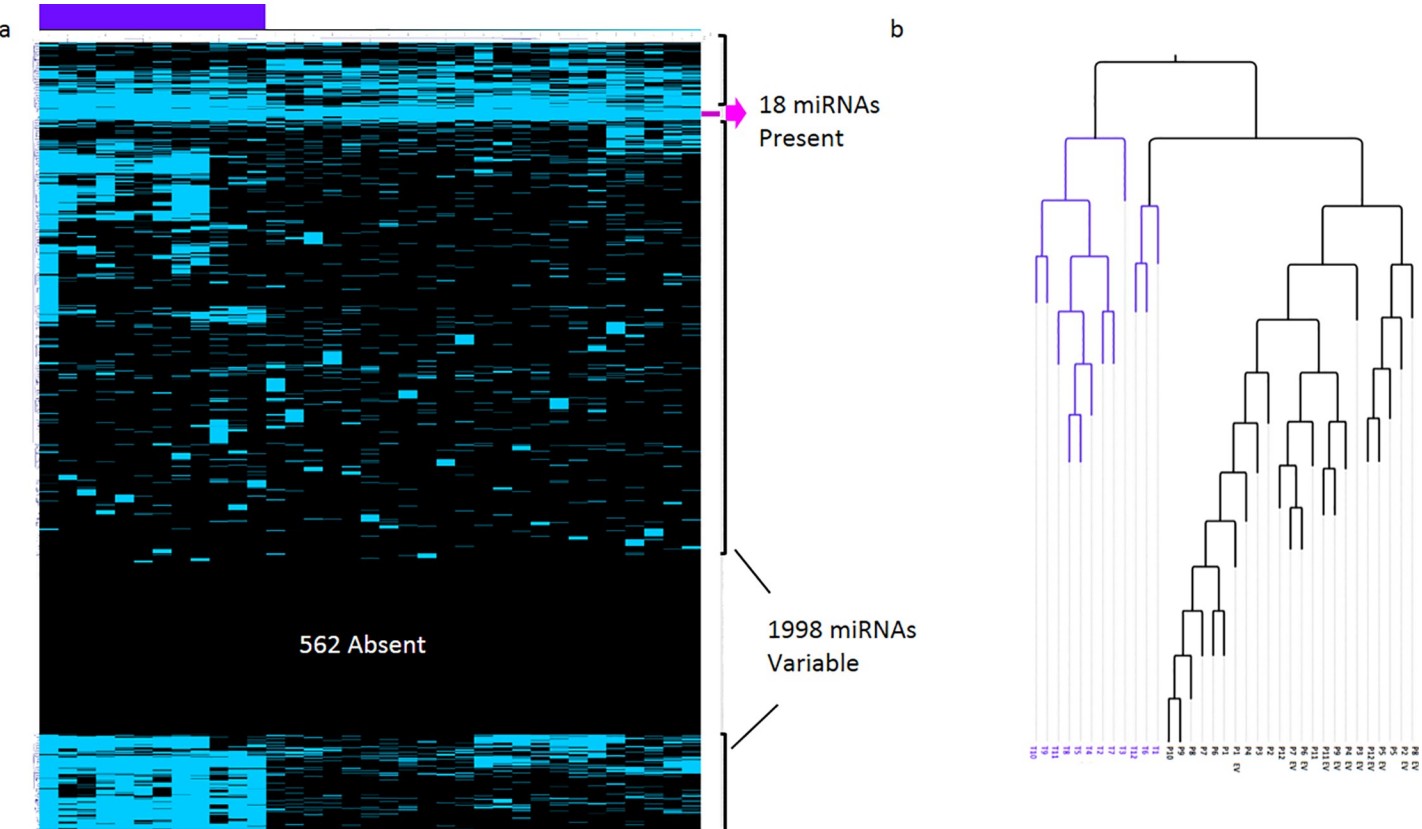

**Fig 1. Mirnomic detection map of 12 primary retinoblastoma, corresponding plasmas and EVs.** a. Heatmap obtained with hierarchical clustering using the detection score. Blue means detected miRNA, while black means undetected miRNA. The columns represent samples while the rows represent the miRNAs (2578). The purple bar at the top left indicates the cluster of the tumor samples. The fuchsia arrow to the right indicates a cluster of 18 miRNAs shared by all samples. There are 1998 variably detected miRNAs, which are split into three miRNA clusters: one at the top, one in the middle, and one at the bottom. b. Corresponding dendrogram: the purple branches indicate tumor samples, while black branches indicate plasma or EVs.

## Mirnomic landscapes in plasma from Rb cases and healthy controls

We used the detection scores from Rb plasma cases to generate a mirnomic map that is shown in **Fig 2A**.

miRNAs detected in plasma from Rb patients are shown in green and those miRNAs not detected are in black. This map shows a mirnomic detection view composed of three types of clusters. First, at the bottom, one cluster with 1435 miRNAs uniformly not detected in all the samples; second, an upper cluster with 1115 miRNAs variably detected across samples; and third, within the upper cluster there is a remarkably small nested cluster of 28 miRNAs shared by all plasma samples from Rb patients. In order to compare this plasma map from Rb cases with those from healthy controls, we generated the same type of mirnomic map with plasma from healthy controls shown in **Fig 2B**. One pair of plasma/EVs from healthy controls yielded unreliable quality controls and was excluded from further analysis. The mirnomic map in plasma from the remaining 11 controls overall shows a structural similarity with that of the Rb cases, but the number of miRNAs in each cluster differs. There is a large cluster of 1372 miR-NAs that are uniformly not detected, a group of 1193 variably detected miRNAs and a smaller nested cluster that are shared by all plasma samples of the healthy controls which is composed of only 13 miRs. The list of miRs contained in each of these clusters can be found in **S1 and S2 Lists**.

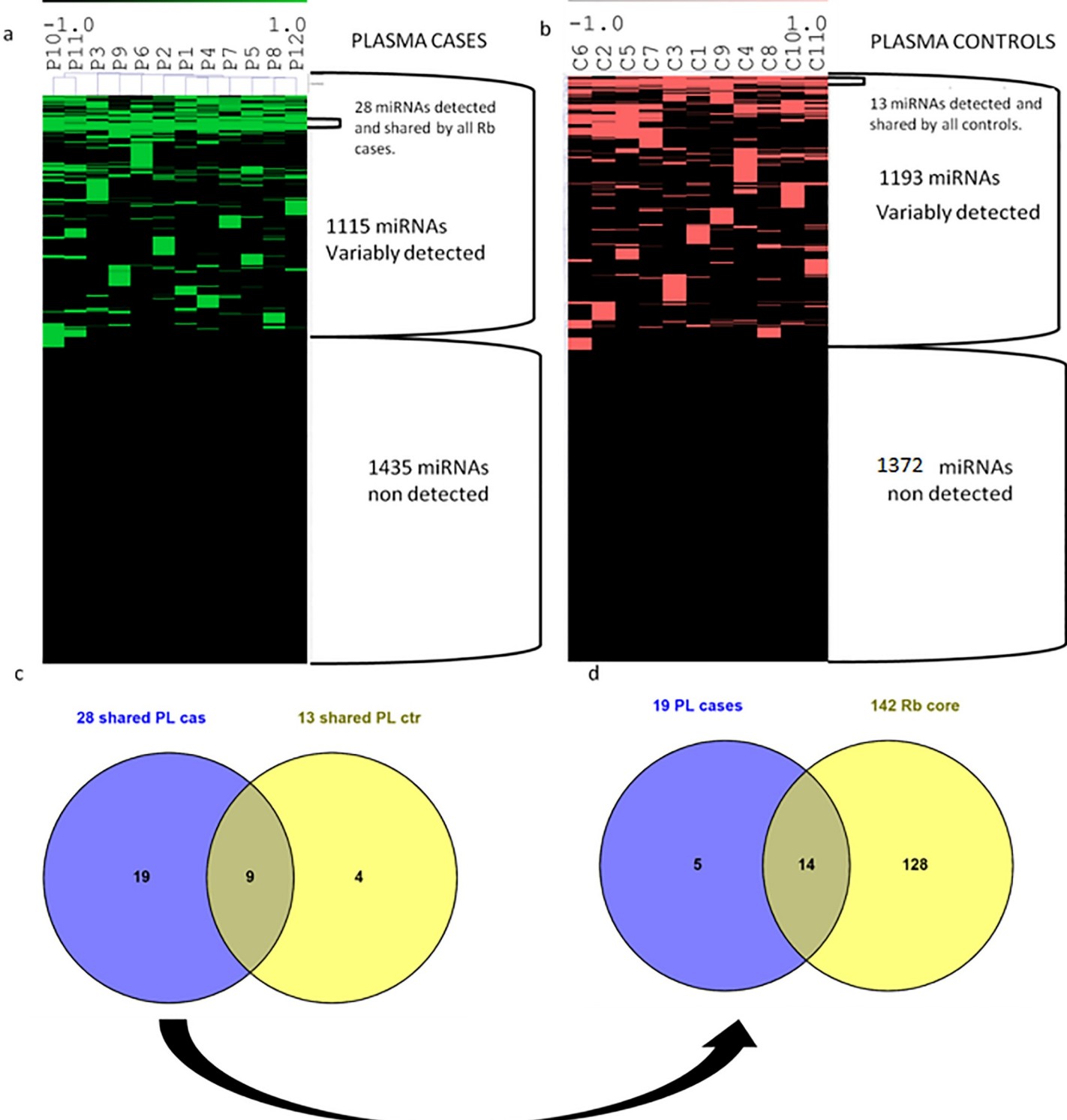

**Fig 2. Analysis from miRNA clusters in plasma shared by all Rb cases and by all healthy controls.** a. Hierarchical clustering of miRNome from Rb cases, showing in green those detected, and in black, those miRNAs not detected. b. Hierarchical clustering of miRNome from healthy controls showing in red those miRNA detected, and in black those not detected. c. Venn diagrams of the miRNAs shared by all Rb cases and by all healthy control showing a group of 9 miRNAs shared by all children and a 19 miRNA group present exclusively in plasma from children with Rb. d. Venn diagrams demonstrating that 14 of the 19 miRNAs exclusive for Rb cases are also present in the 142 miRNA core reported from the corresponding primary tumors [20].

## Comparison of exclusive miRNAs between Rb cases and healthy controls, reveals an Rb signature of 19 miRNAs

We then focused on the small clusters of 28 miRs shared only among all Rb cases and 13 miRs shared only among all healthy controls. We used Venn diagrams in order to test and discriminate if there are corresponding relationships between these sets. As shown in **Fig 2C**, there is a subset of 19 miRNAs exclusive to Rb cases, an intersected subset of 9 miRNAs shared by both cases and controls, and a subset of 4 miRs present exclusively in healthy controls. The corresponding list of these miRNAs including the 19 exclusive miRNAs in plasma from Rb cases is found in **S3 Lists**. We then explored the corresponding relationships of the 19 miRNAs found exclusively in plasma from Rb cases with the central core of 142 miRNAs previously reported in the primary tumors [20]. Results are shown in **Fig 2D**. There is an intersection between 14 out of 19 miRNAs detected in plasma from the Rb cases that are also detected in retinoblastoma tumor tissue, indicating that 5 miRNAs in circulation are not present in the tumor.

## Mirnomic landscapes in EVs from Rb cases and healthy controls

To explore the miRNome in EVs from Rb cases and healthy controls, we generated the same mirnomic detection maps with RNA isolated from EVs. For one Rb case and one healthy control, RNA from the EVs was insufficient for further analysis. In **Fig 3A and 3B** the detection mirnomic maps show also a broad similarity to the maps generated from plasma miRNomes, and again the number of miRNAs in each cluster is different.

Each detection map is composed of a large cluster of non-detected miRNAs, a group of variably detected miRNAs and nested within, small miRNA clusters: a cluster of 36 miRNAs shared in EVs from all Rb cases and a cluster of 30 miRNAs shared in EVs from all controls. The list of miRNAs in these clusters can also be found in **S4 Lists**. An analysis of correspondence between EVs from cases and controls performed with Venn diagrams in **Fig 3C** shows a subset of 13 miRs exclusively detected in EVs from Rb cases, an overlapping subset of 23 miRs shared by both cases and controls, and a subset of 7 miRNAs detected exclusively in EVs from healthy controls (**S5 List**). **Table 2** shows a summary of all these clusters and the number of miRNAs belonging to each one.

In bold type are the highest number of miRNAs for each class of clusters in plasma and EVs: not-detected in all, variably detected in all, shared detected in all.

To explore how many of the 13 miRNAs that were exclusive to EV from cases were also detected in the primary tumor, we also performed a correspondence test against the central 142 miRNA core using Venn diagrams, and found that 8 out of these 13 miRNAs were also detected in the primary tumor as shown in **Fig 3D**.

## Differential analysis performed poorly between Rb cases and healthy controls

With two putative Rb case signatures, one with 19 miRs in plasma and one with 13 miRs in EVs, we subsequently asked if differential analysis using the detection levels would also be able to identify these signatures. Using Limma [26] in R from Bioconductor and setting a $p$ value $< 0.05$, we performed differential analysis in miRNAs between Rb and healthy controls first among plasma samples and then among EVs. In the differential analysis, we found 158 differentially expressed miRNAs in plasma and 142 differentially expressed miRNAs in EVs. However, adjusted $p$ $values$ indicate that differences were not significant for any of them. Only 5 miRNAs from the signatures were overexpressed in Rb: 3 of these were detected in plasma and 2 in EVs. The highest differentially expressed miRNAs are shown in Tables 3 and 4.

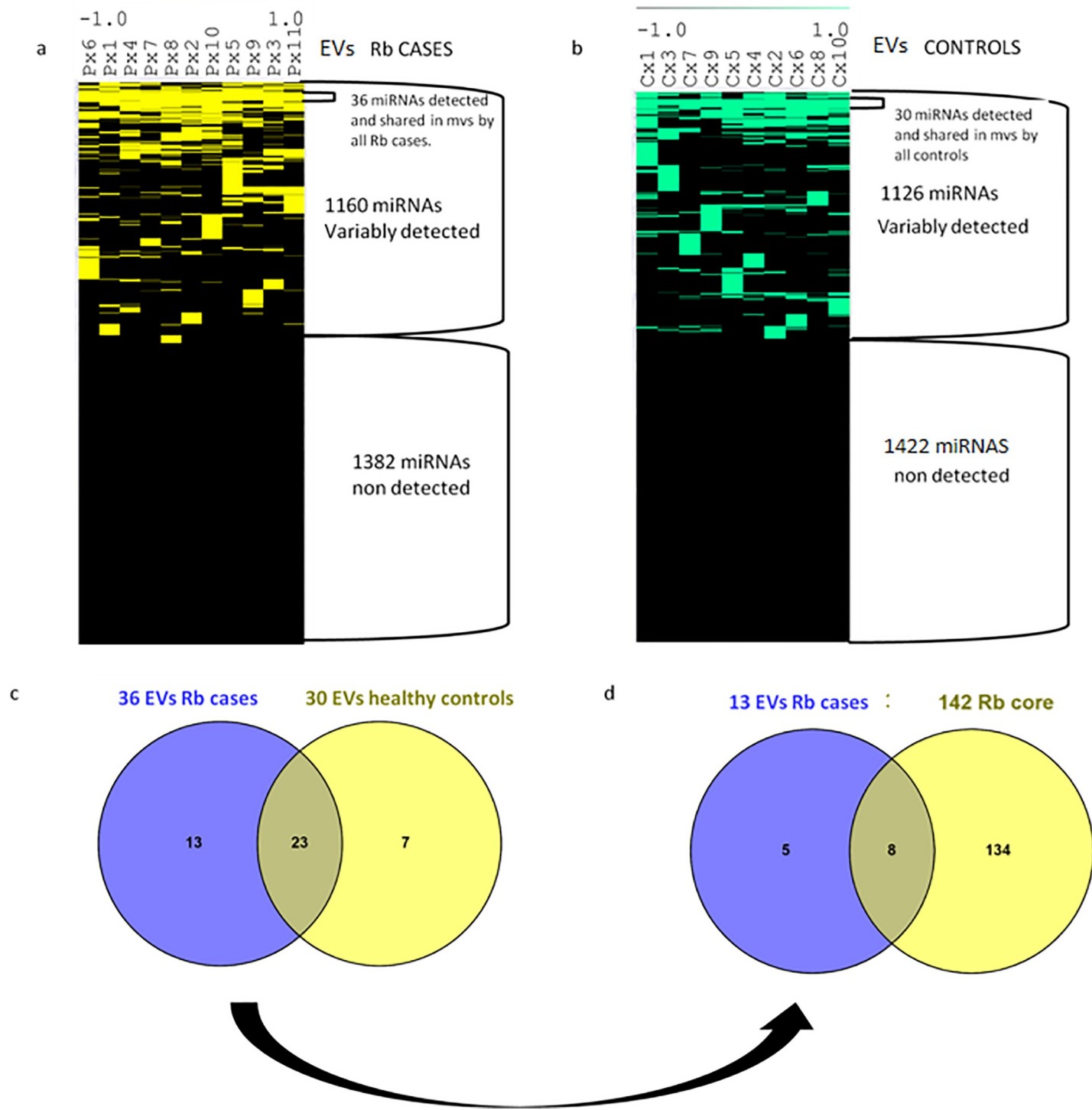

**Fig 3. Analysis from miRNA clusters in EV shared by all Rb cases and by all healthy controls.** a. Hierarchical clustering of miRNome from EVs/Rb cases showing in yellow those detected, and in black, those not detected. b. Hierarchical clustering of miRNome from EVs controls showing in turquoise blue those detected, and in black, those not detected. c.Comparison using Venn diagrams of miRNAs shared in EVs from all cases with the miRNAs shared in EVs from all controls yields a 13 miRNA group present only in Rb cases. d.From this 13 miRNA group shared only among Rb cases, only 8 are also detected in the 142 miRNA core of the primary tumor.

**Table 2. Number of miRNAs in clusters found in plasma and EVs miRNome from Rb cases and healthy controls.**

| miRNome ID | Shared not-detected miRNAs | Variably detected miRNAs | Shared miRNAs |
|---|---|---|---|
| Rb Cases Plasma | **1435** | 1115 | 28 |
| Rb Cases EVs | 1409 | 1137 | **32** |
| Retinoblastoma tissues | 995 | 1441 | 142 |
| Healthy Controls | 1317 | **1248** | 13 |
| Healthy Controls EVs | 1378 | 1170 | 30 |

The cells shaded in yellow correspond to those miRNAs that were also identified by our alternate method in the cluster of shared miRNAs, in Table 2 our proposed signature.

The corresponding heatmap does not show color differences among cases and controls except for the 158 and 142 differentially expressed miRNAs, which are shown in **Fig 4**. Even though differential analysis identified miRNAs with p-values <0.05, the adjusted p-values did not show any statistically significant differences between Rb cases and controls for any miRNA not in plasma not in EVs.

## Plasma signature of 19 miRNAs discriminates correctly Rb cases from healthy controls

Since 19 miRNAs as a group or signature are detected in plasma from all Rb cases, and 13 miRNAs as a signature are detected in EVs from Rb cases, we plotted the profile detection scores against controls to test how well the signatures were able to distinguish both groups. The discriminating ability of the 19 miRNAs plasma profile is illustrated in **Fig 5A**. Importantly, not a single control had the 19 miRNA signature.

In contrast, the 13 miRNA profile from EVs cannot distinguish both classes, as two healthy controls display the 13 miRNA signature corresponding to Rb patients. This indicates that the more robust circulating Rb case signature is in the plasma compartment not in EVs.

In order to evaluate the abundance of the 19 miRNA circulating signature, we shifted to analysis of detection levels. We plotted in histograms the average signal for each miRNA across the plasma dataset obtained from Rb cases. **Fig 5B** shows that all signals are well above the general average. In an attempt to compact the 19 miRNA signature found in Rb plasmas, we tested the correspondence between the 19 and the 13 miRNA signature found in Rb plasma and EVs using Venn diagrams. Even though we found an overlapping set of 7 miRNAs present in both compartments, the discriminating power of this 7miRNA 'signature' performed worse than the parental 13 signature found in EVs, and was not able to distinguish 5 individuals as controls. A summary of shared clusters and intersected results with Venn diagrams described here are illustrated in **Fig 5C**.

## The detection mirnomic scores reveal the size of the circulating miRNome

An important application of the detection score is the possibility to describe and summarize the number of detected miRNAs per sample, per group (cases or controls) and per class (all the children in this study). The results of this exploration are presented in **Table 5**.

**Table 3. The most differentially expressed miRNAs in Rb cases.**

| miRNA | logFC | AveExpr | t | *p value* | adj. *p val* |
|---|---|---|---|---|---|
| hsa-miR-3613-3p | 2.852 | 7.792 | 2.675 | 0.013 | 0.808 |
| hsa-miR-4529-3p | 2.803 | 5.069 | 2.356 | 0.027 | 0.833 |
| hsa-miR-4668-5p | 2.087 | 7.569 | 2.159 | 0.041 | 0.833 |
| hsa-miR-4508 | 1.345 | 3.287 | 3.397 | 0.002 | 0.667 |

**Table 4. The most differentially expressed miRNAs in EVs Rb cases.**

| EVs Rb cases | | | | | |
|---|---|---|---|---|---|
| miRNAs | logFC | AveExpr | t | *p value* | adj. *p val* |
| hsa-miR-4529-3p | 2.656 | 4.323 | 3.074 | 0.006 | 0.784 |
| hsa-miR-6800-3p | 0.937 | 2.958 | 2.189 | 0.04 | 0.784 |
| hsa-miR-3921 | 0.797 | 2.919 | 2.566 | 0.018 | 0.784 |

The number of miRNAs detected in plasma is approximately the same for cases and controls, with an average of 534.7 miRNAs for plasma cases and 540 for plasma controls. In contrast, the average number of miRNAs detected in EVs from cases, 656.2, is higher than the average number of miRNAs detected in EVs from controls, 596. Thus, we found 60 miRNAs more in EVs from cases than in controls. The average size of the circulating miRNome in plasma for these children is 537 miRNAs, while the miRNome size in EVs is an average of 625 miRNAs, with an overall difference of 88 more miRNAs in EVs than in plasma.

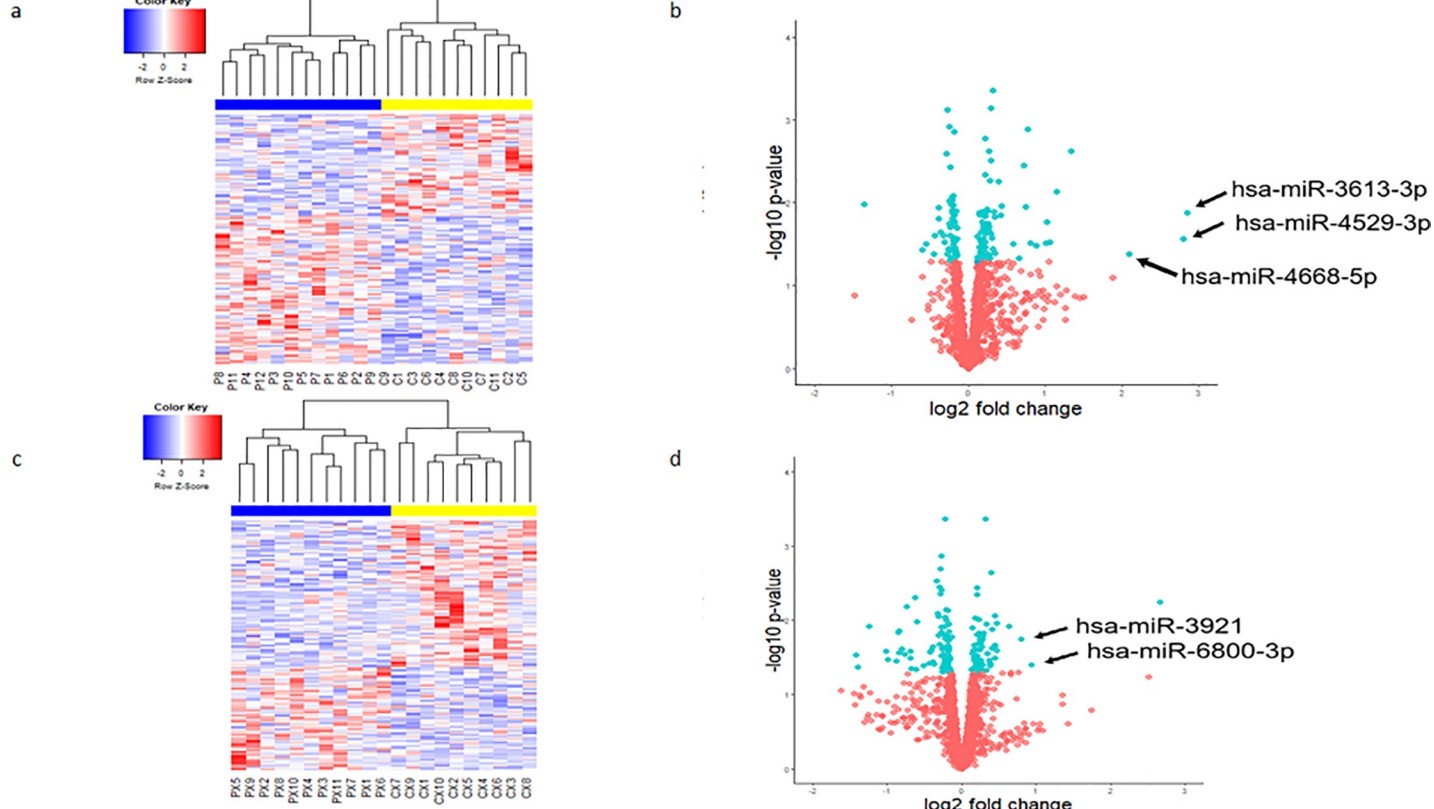

**Fig 4. Differential expression analysis using detection levels.** a. A heatmap showing the 158 most differentially expressed miRNAs in plasma. miRNAs with higher detection levels are represented in red, while miRNAs with lower detection levels are in blue, and those with no significant difference are shown in white. The blue and yellow bar on top of the figure indicates samples from Rb cases and samples from healthy controls respectively. b. Volcano plots illustrate fold change (log base 2) compared with *p* value (−log base 10). The three labeled miRNAs were those that were also identified in the cluster of miRNAs shared in the plasma from all Rb cases. c. A Heatmap showing the top 142 differentially expressed miRNAs in EVs. d. Indicated miRNAs also identified in the cluster of shared miRNAs in EVs from all Rb cases.

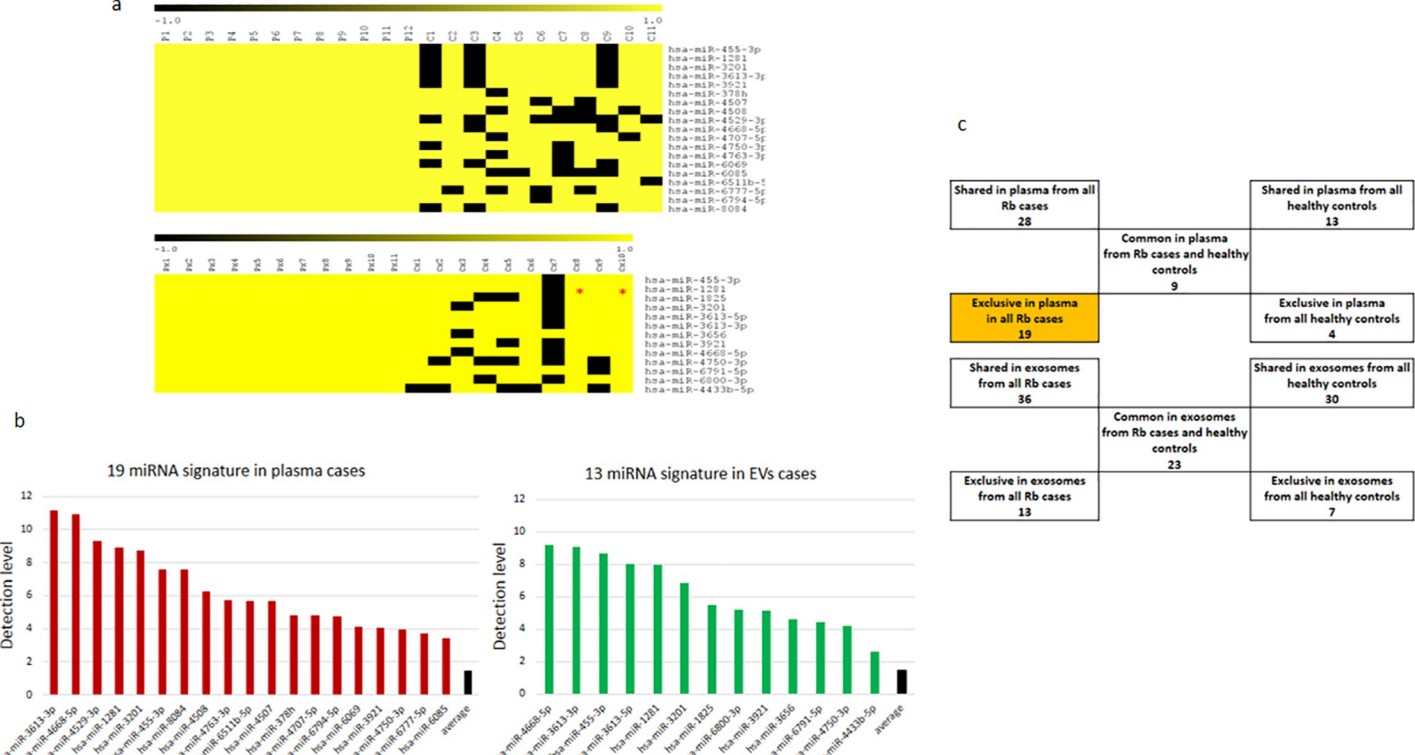

**Fig 5. Detection score maps of Rb signatures against healthy controls.** a. The 19 miRNAs signature consistently detected in plasma from all Rb cases distinguish correctly all controls. The 13 miRNA signature consistently detected in EVs from Rb cases cannot distinguish correctly all controls, red asterisks indicate those healthy controls with EVs Rb profiles. b. Detection levels of the 19 and 13 Rb signatures found in plasma and EVs respectively are above the average. c. Summary of shared and exclusive clusters found among Rb cases and healthy controls. The dark yellow box highlights the cluster able to discriminate Rb cases from all healthy controls.

## Common miRNAs shared by Rb cases and healthy controls may be useful as references or normalizers

Considering that the methodological approach presented here, can identify distinctive and unbiased miRNAs among groups or classes, with the use of the detection score, we explored whether detectable miRNAs shared across all the samples could identify potential endogenous normalizers. To do this we generated a dataset with detection scores from every sample (plasma and EVs) in this study and performed a hierarchical clustering analysis. As shown in

**Table 5. Plasma and EVs miRNome size described using detected miRNAs in Rb cases and healthy controls.**

|  | Mean | Minimum | Maximum |
|---|---|---|---|
| miRNAs in plasma* | 537 | 398 | 690 |
| miRNAs in EVs* | **625.2** | 452 | 846 |
| miRNAs in plasma from Rb cases | 534.7 | 454 | 690 |
| miRNAs in plasma from healthy controls | **540** | 398 | 634 |
| miRNAs in EVs from Rb cases | **656.2** | 506 | 846 |
| miRNAs in EVs from healthy controls | 596.9 | 452 | 756 |

*Cases and controls combined

Bolded text signals the highest number of miRNAs analyzed from each compartment.

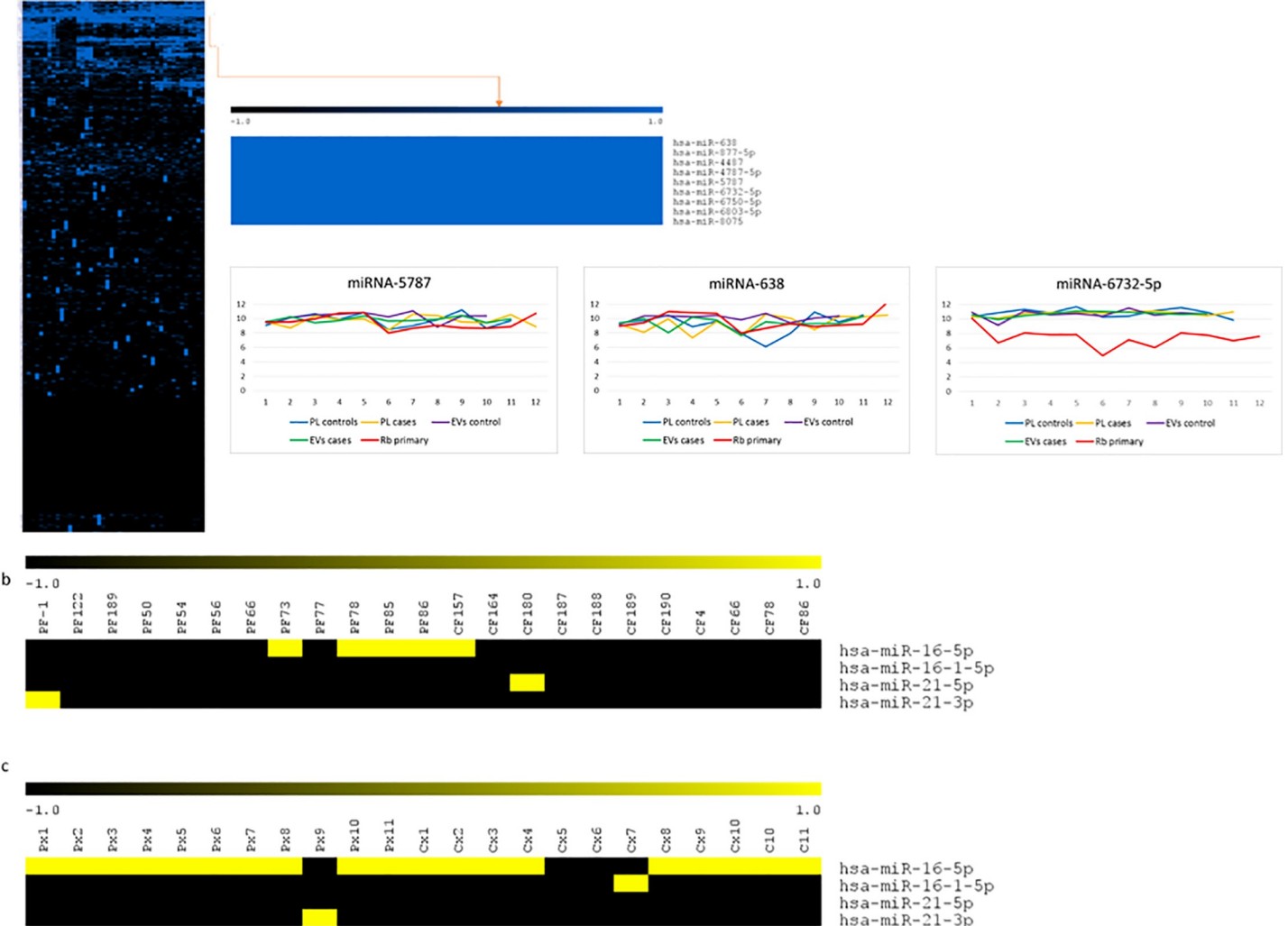

**Fig 6. Detected miRNAs across all samples.** a. Hierarchical clustering with all datasets used in this study identified a 9 miRNA cluster detected in all samples. Variance across groups including the primary tumors is shown for the three miRNAs that had lower variance. b. The Detection score in plasma for those miRNAs commonly used as normalizers. c. The Detection score in EVs for the same miRNAs.

**Fig 6A**, there is indeed a small group of 9 miRNAs that is consistently detectable in every sample. This constitutes a relevant intergroup stability feature.

Next, to determine how much variability these miRNAs had across the whole group, we shifted the analysis to the corresponding detection levels and calculated the variance for each of these 9 miRNAs. **Table 6** shows that miRNA-5787, miR-638 and miRNA-6732-5p had the lowest variance within our dataset.

The plotted variance data of these three miRNAs is shown in **Fig 6A**, and is disaggregated by blood compartment (plasma/EVs) and by group (Rb cases/controls) including data from the primary tumors. This results show considerable inter and intragroup stability. Additionally the high detection levels of these three miRNAs suggest that they are potential endogenous plasma normalizers.

To contrast the detection score and variance analysis of these potential endogenous normalizers with some normalizers currently used in plasma, we plotted detection scores of miR-16, miR-21 (common normalizers used in qRT-PCR) across all samples in **Fig 6B**. It was

**Table 6. Variance analysis and detection level of new potential blood miRNAs normalizers miRNAs detected in all samples.**

| miRNAs ID | Variance | Detection level Average | Detection level Median |
|---|---|---|---|
| hsa-miR-5787 | 0.60 | 9.71 | 10.40 |
| hsa-miR-638 | 1.19 | 9.78 | 9.82 |
| hsa-miR-6732-5p | 2.31 | 4.24 | 3.77 |
| hsa-miR-6803-5p | 3.02 | 6.22 | 6.56 |
| hsa-miR-8075 | 3.25 | 6.23 | 6.45 |
| hsa-miR-877-5p | 3.41 | 9.52 | 9.57 |
| hsa-miR-4787-5p | 4.34 | 8.20 | 8.70 |
| hsa-miR-4487 | 4.98 | 9.97 | 10.56 |
| hsa-miR-6750-5p | 5.77 | 8.94 | 9.43 |

interesting to note that miR-16-5p signals were detectable only in 17 of 21 samples, and were mainly present in EVs. In contrast, miR-21 is detected only in a few samples. Consistent with this detection data, we show the variance analysis and detection levels of these two miRNAs at the bottom of **Table 7**. It is important to note that detection levels are rather low.

## Mirnomic analysis by sex

Further exploration with this analytical approach led us to considering participant's sex and test if the sex of individuals in the whole group could allow us to find associated miRNAs. Results in **Fig 7** show that all females shared 14 miRNAs while all males shared 13 miRNAs. Venn diagrams show the 9 overlapping miRNAs are those analyzed in the previous section, while 5 miRNAs were detected exclusively in females and 4 miRNAs were detected exclusively in males (**S6 Lists**).

A further search for miRNA targets as well as pathway analysis of these sex associated miRNAs using miRPathv3, TargetScan and TarBase v7, indicated that some of these miRNA targets and pathways were related to estrogen and thyroid signaling pathways in females and steroid biosynthesis in males among others.

## Discussion

As part of an international and multi institutional study to determine risk factors for retinoblastoma [22], we created a bio repository of frozen plasma from children with Rb and healthy controls. In a group of those plasma samples, we tested whether miRNA profiles are able to discriminate Rb cases from healthy controls. This work is an extension of a previous effort to define the mirnomic view in primary retinoblastoma tissue [20]. We now define the mirnomic view of the corresponding plasmas, further comparing them with healthy controls.

We propose a novel approach for 'signal' treatment that simplifies and enables the discovery of meaningful miRNA data patterns in two blood compartments, total plasma and EVs. In the

**Table 7. Variance analysis and detection levels of common normalizers in use miRNAs commonly used as qRT-PCR normalizers in blood.**

| miRNAs ID | Variance | Detection level Average | Detection level Median |
|---|---|---|---|
| hsa-miR-21-5p | 0.67 | 1.38 | 1.22 |
| hsa-miR-21-3p | 0.02 | 1.26 | 1.26 |
| hsa-miR-16-5p | 15.89 | 4.24 | 1.57 |
| hsa-miR-16-1-3p | 0.05 | 1.23 | 1.21 |

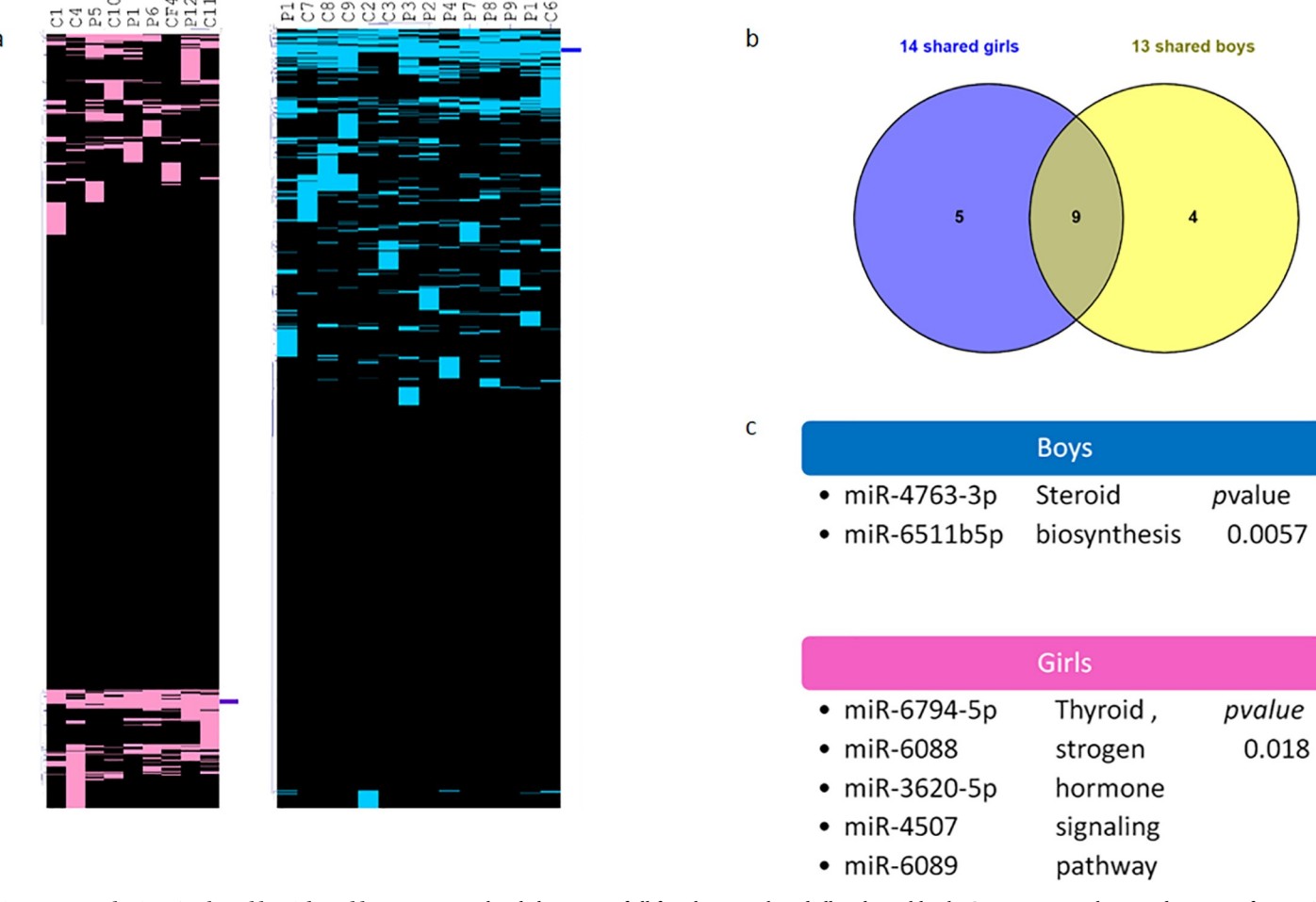

**Fig 7. Detected miRNAs shared by girls and boys.** a. Hierarchical clustering of all females in pink and all males in blue b. Comparison with Venn diagrams of miRNAs shared by all females and miRNAs shared by all males. c. related functional pathways of 3 female specific and 2 male specific miRNAs.

search for biologically informative and clinically meaningful biomarkers for cancer, it is important to know which information can be found in the different blood compartments. Our finding that miRNA contents in total plasma and EVs are distinct may be particularly informative. One possible explanation may be that the purification of EVs may amplify signals from miRNAs that may not be detectable in miRNAs extracted from whole plasma. Importantly, the 19 miRNA signature that is able to discriminate Rb cases from healthy controls is found only in RNA from total plasma and not in EVs, this is relevant because the method to obtain RNA from plasma is simpler and less expensive than the method to obtain RNA from of EVs.

We opted to use the concept of "detection" levels instead of "expression" levels, because plasma is essentially an extracellular biofluid, EVs are not cells, and 'expression level' refers to an intracellular biological phenomenon.

An important measurement that detection scores can provide is the miRNome size. In fact, size of the circulating miRNome has not previously been reported. Even though the number of samples analyzed is too small to permit examination of statistical significance, the information we report constitutes firm steps towards establishing the biological meaning of these differences.

One of the motivations to do this study is the remarkable stability reported for miRNAs in plasma. In order to assure reproducibility the samples have been collected and processed

under standard procedures. Our analytic approach focused first on generating the detection score that simplifies microarray data, and later on examination of expression measurements. In our pilot work, we performed technical replicas, and since both expression and detection data were highly correlated, we opted to prioritize and include more biological replicas in our design instead of additional technical replicas, to increase the possibilities of finding patterns shared by Rb cases. Expression measurements in microarray platforms are very sensitive and detect large amounts of 'noise', we believe that this sensitivity masks underlying data patterns, which are unmasked by using the detection score. This noise reduction in the signals facilitates the recognition of patterns in the data when using unsupervised algorithms.

Using clinical information as guides to find molecular associations, allowed the discovery of regularities in the data and molecular patterns associated to Rb cases, associated to healthy controls, associated to all samples, and associated to male or to females. The use of clinical descriptors in combination with detection score analysis is the key methodology that allowed us to discover, more accurate biological molecular relationships and to better interpret the data.

In the plasma from children with Rb, we discovered a group of 19 miRNAs shared by all Rb cases that are present as a profile or molecular signature in all our Rb cases and not in healthy controls. In this signature, there are 14 miRNAs also detected in the corresponding primary tumor, presumably a circulating tumor signature, and it is interesting that 5 miRNAs in this signature are not detected in the tumor miRNA core. Even though there are more miRNAs in EVs than in plasma, it is remarkable that the most robust pattern able to discriminate cases from controls is not present in EVs and is present in total plasma. At this moment, we can only speculate about why the plasma signature clearly better discriminates Rb cases from healthy controls compared with the EVs signature. One possibility is that detectable miRNAs in plasma might reflect both the circulating tumor message as well as a systemic response to cancer.

Differential analysis found miRNAs that significantly differentiated Rb case from those of controls, though none remained statistically significant after adjustment. This indicates that individual miRNA differences between Rb cases and controls are not very strong. Instead, this suggests that useful and robust miRNA differences will more likely be based on a combination of larger miRNA groups or patterns.

We also discovered a small cluster of 9 miRNAs shared across all the samples, the low variance across the samples supports its potential use as a circulating normalizers. Our results add precision and expand information on miR-16-5p and miR-21 common normalizers for qRT-PCR in plasma, since we show that the blood compartment on which miR-16-5p is detected in most samples are EVs. We also show and contrast average detection levels as well as variability measurements (variance) across the samples under study. The expression levels between these common normalizers and the three most highly expressed newly proposed normalizers (miR-5787, 638 and 6732-5p) show that the later have higher detection levels and justify to continue the experimental exploration for using them as normalizers.

Our approach affords a glimpse into parameters for sensibility and specificity that can be used in the development of useful biomarkers. A limitation is that we were only able to partially validate a few miRNAs with qRT-PCR. One potential explanation for this may be an enzymatic amplification bias suggesting a need for better methods for miRNA detection and quantification.

## Conclusions

The possibility to study components of malignant solid tumors in blood, whether DNA, RNA or EVs, in other words, the so-called 'liquid biopsy' is a very active and promising field of

research. Despite the great interest in this topic, the field itself is still very young, and there are many unknowns in almost every aspect of this field. This study offers a practical approach for navigating and exploring high throughput mirnomic profile data, using a combination of clinical information and unsupervised computational tools in order to reach integrative and plausible interpretations, thus opening up the possibility of differentiating normal and 'not' normal mirnomic profiles.

## Supporting information

**S1 Lists. Mirnomic clusters in Rb plasma.**
(XLSX)

**S2 Lists. Mirnomic clusters in healthy controls.**
(XLSX)

**S3 Lists. 19 exclusive miRNAs in Rb plasma.**
(XLSX)

**S4 Lists. Mirnomic clusters in EVS from Rb cases.**
(XLSX)

**S5 Lists. Mirnomic cluster in EVS from healthy controls.**
(XLSX)

**S6 Lists. miRNA clusters by sex.**
(XLSX)

## Acknowledgments

We thank Josefina Romero Rendón [TS] for outstanding support during families' recruitment.

## Author Contributions

**Conceptualization:** Noé Durán-Figueroa, M. Verónica Ponce-Castañeda.

**Data curation:** Blanca Elena Castro-Magdonel, Diana E. Alvarez-Suarez, M. Verónica Ponce-Castañeda.

**Formal analysis:** Blanca Elena Castro-Magdonel, Diana E. Alvarez-Suarez, Noé Durán-Figueroa, María de Jesús Orozco-Romero, M. Verónica Ponce-Castañeda.

**Funding acquisition:** Manuela Orjuela, Noé Durán-Figueroa, M. Verónica Ponce-Castañeda.

**Investigation:** Blanca Elena Castro-Magdonel, Lourdes Cabrera-Muñoz, Citlali Lara-Molina, Daphne García-Vega, Noé Durán-Figueroa, Adriana Hernández-Ángeles, M. Verónica Ponce-Castañeda.

**Methodology:** Blanca Elena Castro-Magdonel, Noé Durán-Figueroa, María de Jesús Orozco-Romero, M. Verónica Ponce-Castañeda.

**Project administration:** Adriana Hernández-Ángeles, M. Verónica Ponce-Castañeda.

**Resources:** Manuela Orjuela, Lourdes Cabrera-Muñoz, Stanislaw Sadowinski-Pine, Aurora Medina-Sanson, Citlali Lara-Molina, Daphne García-Vega, Yolanda Vázquez, Noé Durán-Figueroa, M. Verónica Ponce-Castañeda.

**Software:** Diana E. Alvarez-Suarez.

**Supervision:** Javier Camacho, Lourdes Cabrera-Muñoz, Stanislaw Sadowinski-Pine, Noé Durán-Figueroa, M. Verónica Ponce-Castañeda.

**Visualization:** Blanca Elena Castro-Magdonel, Diana E. Alvarez-Suarez.

**Writing – original draft:** M. Verónica Ponce-Castañeda.

**Writing – review & editing:** Blanca Elena Castro-Magdonel, Manuela Orjuela, Javier Camacho, Aurora Medina-Sanson, Citlali Lara-Molina, Noé Durán-Figueroa, M. Verónica Ponce-Castañeda.

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
