## [Decision Letter · Decision Letter 0]

10 Jan 2020

PONE-D-19-34857

miRNome landscape analysis in plasma and corresponding extracellular vesicles from retinoblastoma patients and healthy controls reveals a circulating tumor signature, the size of the circulating miRNome and new normalizers candidates

PLOS ONE

Dear Dr. Ponce-Castañeda

Thank you for submitting your manuscript to PLOS ONE. After careful consideration, we feel that it has merit but does not fully meet PLOS ONE’s publication criteria as it currently stands. Therefore, we invite you to submit a revised version of the manuscript that addresses the points raised during the review process.

We would appreciate receiving your revised manuscript by Feb 24 2020 11:59PM. To enhance the reproducibility of your results, we recommend that if applicable you deposit your laboratory protocols in protocols.io, where a protocol can be assigned its own identifier (DOI) such that it can be cited independently in the future. For instructions see: http://journals.plos.org/plosone/s/submission-guidelines#loc-laboratory-protocols

We look forward to receiving your revised manuscript.

Kind regards,

Zhijing Tan, Ph.D

Academic Editor

PLOS ONE

2. Please note that all PLOS journals ask authors to adhere to our policies for sharing of data and materials: https://journals.plos.org/plosone/s/data-availability. According to PLOS ONE’s Data Availability policy, we require that the minimal dataset underlying results reported in the submission must be made immediately and freely available at the time of publication. As such, please remove any instances of 'unpublished data' or 'data not shown' in your manuscript and replace these with either the relevant data (in the form of additional figures, tables or descriptive text, as appropriate), a citation to where the data can be found, or remove altogether any statements supported by data not presented in the manuscript.

Additional Editor Comments (if provided):

I would be pleased to recommend further for publication a revised manuscript that addresses all reviewers' concerns.

Reviewers' comments:

Reviewer's Responses to Questions

**Comments to the Author**

1. Is the manuscript technically sound, and do the data support the conclusions?

Reviewer #1: Yes

Reviewer #2: No

2. Has the statistical analysis been performed appropriately and rigorously? 

Reviewer #1: Yes

Reviewer #2: Yes

3. Have the authors made all data underlying the findings in their manuscript fully available?

Reviewer #1: Yes

Reviewer #2: Yes

4. Is the manuscript presented in an intelligible fashion and written in standard English?

Reviewer #1: Yes

Reviewer #2: No

5. Review Comments to the Author

Reviewer #1: Castro-Magdonel and colleagues carried out a survey on miRNAs from retinoblastoma children’s plasma or extracellular vesicles and compared the miRNAs patterns with healthily age matched children, which provided us an informative resource of the landscape analysis of miRNAs in retinoblastoma children.

However, there are some concerns should be addressed before consideration for publication.

The title of the manuscript is too long. Modifications required.

Line 35: Circulating miRNAs are potential useful biomarkers in medicine.

Why they are only important in medicine? The authors should include all the related fields miRNAs involved.

Line 39: To address these questions,

It seemed that the authors did not mention any questions or issues regarding miRNAs in previous couples’ sentences. Clarify them.

Line 67: In plasma have been found both, in soluble form associated to Argonaut proteins, and also inside extracellular vesicles or exosomes (6)(7).

This sentence is hard and difficult to understand. Make it readable.

Line 88 to Line 103:

The authors attempted to make their methodology more convincing, logic, and fit their study very well, but they failed. Recommends revision and rearrangements to make it concise and precise.

Materials and methods

The controls and the Rb patients’ age and sex are not matched well. Also, the body weight or BMI, and race should be included if it is not unavailable to the authors.

Further, in microarray experiments, variations of expression measurements can be attributed to many different sources that influence the stability and reproducibility of microarray platforms. How the authors handle this issue? Is normalization applied when process the data?

Results and discussions

There are several studies reported about 1000 to 2000 miRNAs in Rb patients’ plasma (https://www.ncbi.nlm.nih.gov/pmc/articles/PMC5493862/). Why the authors only detected about 600 miRNAs in plasma?

How the authors validate the miRNA expression profile? Any qRT-PCR validation applied?

The figures’ resolution is always low, which are difficult to read.

Reviewer #2: In this paper, the authors try to answer the questions what the differences are between the healthy person and Rb (diseased) person in terms of their miRNAs in plasma and extracellular vesicles using 12 patients and 12 controls. The size of miRNAs in plasma are 537 and in EV is 626. And they found that 19 miRNAs in plasma are significant between healthy and diseased person. And they also found 9 miRNAs that can be detected in all the samples, suggesting that these 9 miRNAs can be used as internal control. They also found that the gender is also a factor that affects the miRNAs between healthy and diseased person. First, I think the English in this paper needs to be improved. Second, I think using 12 patients and 12 control as samples are too few to get conclusion. Third, the difference between the plasmid and EV in terms of its chemical components should be discussed in the paper. Other than that, I have some comments as below:

1. Methods/Patients and samples: What are including criteria and what are excluding criteria?

2. How are the healthy control being selected?

3. How many times the experiments for Fig 1, 2, and 3 are done? If they are only done once, how do the authors make sure the results are reliable?

4. In figure 3 and 4, how top is the top ranked miRNAs? Are the top miRNAs showed in Table 3 and Table 4 same as those detected by mirnomic analysis?

5. In Figure 5 and table 5, the authors summarized the number of detected miRNAs per sample, per group (cases or controls) and per class. How does these numbers compare to other mirnome studies?

6. Figure 7 indicated that gender may be a factor that cause the miRNA to be different. So Could the authors analyze the difference of miRNA in plasma and EV between diseased person and health person with gender differences? So there will be four groups: 1) health female, 2) health male, 3) diseased female, and 4) diseased male.

7. The authors found that 19 miRNAs shared exclusively by all Rb cases. Are these 19 miRNAs have any similarities? Which pathways did these 19 miRNAs targeted? How about this result compared with other similar results? Is there any difference? If yes, why?

8. Why “miR-16-5p signals are detectable in 17 / 21 samples mainly in EVs” but not all the samples in EVs? And why “miR-21 is detected only in a few samples” but not all the samples? Since miR-16 and miR-21 are all normalizers, they should be seen in all the samples?

9. Line 319: “13 miRNAs as a signature are exclusively detected in EVs from Rb cases.” Please delete exclusively because it is not exclusive as 12miRNAs are also seen in healthy controls.

10. Why “more robust circulating Rb case signature is in the plasma compartment not in EVs”? Could the authors give an explanation?

11. Line 302-303: “the differences are found not too strong” is too subjective. Please revise this sentence, e.g., adding a reference.

12. Table 1: Did the authors do the statistical analysis on the patient gender between the health and diseased group?

13. EVs includes exosomes and microvesicles. In the introduction, the authors only introduce the exosomes but not the microvesicles. Please add some information of microvesicles.

6. PLOS authors have the option to publish the peer review history of their article (what does this mean?). If published, this will include your full peer review and any attached files.

Reviewer #1: No

Reviewer #2: No

---

## [Author Response · Author response to Decision Letter 0]

25 Feb 2020

Reviewer #1: Castro-Magdonel and colleagues carried out a survey on miRNAs from retinoblastoma children’s plasma or extracellular vesicles and compared the miRNAs patterns with healthily age matched children, which provided us an informative resource of the landscape analysis of miRNAs in retinoblastoma children. However, there are some concerns should be addressed before consideration for publication.

The title of the manuscript is too long. Modifications required.

Response= We modified the manuscript title from a 30 word (233 characters) to 22 words (137 characters): 

Plasma miRNome analysis from children with retinoblastoma reveals the size of the circulating miRNome, a tumor signature and novel candidates for normalizers.

Line 35: Circulating miRNAs are potential useful biomarkers in medicine.

Why they are only important in medicine? The authors should include all the related fields miRNAs involved.

R= we agree that miRNAs are important in many fields beyond medicine, to indicate this we included the following in the abstract’s opening statement in the abstract: 

“miRNAs regulate post-transcriptional gene expression in metazoans, and thus are involved in many fundamental cellular biological processes. Extracellular miRNAs are also found in most human biofluids including plasma. These circulating miRNAs constitute a long distance inter cellular communication system and are potentially useful biomarkers.”

Line 39: To address these questions,

It seemed that the authors did not mention any questions or issues regarding miRNAs in previous couples’ sentences. Clarify them.

R= we rearranged the text as follows: 

We proposed to answer how many and which miRNAs are detectable in plasma or extracellular vesicles as these questions have not yet been answered. We set out to address this knowledge gap by analyzing the mirRNome in plasma and corresponding extracellular vesicles (EVs) from 12 children….

Line 67: In plasma have been found both, in soluble form associated to Argonaut proteins, and also inside extracellular vesicles or exosomes (6)(7). This sentence is hard and difficult to understand. Make it readable. 

R= we agree and have modified the sentence as follows:

“In plasma, miRNAs associate with Argonaut proteins and can also be detected inside extracellular vesicles or exosomes (6)(7).” 

Line 88 to Line 103:

The authors attempted to make their methodology more convincing, logic, and fit their study very well, but they failed. Recommends revision and rearrangements to make it concise and precise. 

R= we understand the reviewer’s suggestion and have modified and rearranged the paragraphs as follows:

“High throughput expression microarray technology can scan complete miRNnomes or transcriptomes in a single hybridization experiment. The original technology used a double channel platform and yielded relative measurements. However, more mature technology with one channel platforms include calibrators which yield absolute measurements. These calibrators allowed the development of robust methods for detection calls (19) which are basically ignored by most analysis. Broad maps of what is and is not detected can now be generated with the detection score using clustering tools thus facilitating data interpretation (20). This approach allows answering relevant questions such as ‘how many’ and ‘which’ miRNAs can be detected in plasma or EVs? Similarly, one can now posit: ‘Are detectable miRNAs shared by all samples, phenotypes or experimental conditions?’ 

Materials and methods

The controls and the Rb patients’ age and sex are not matched well. Also, the body weight or BMI, and race should be included if it is not unavailable to the authors.

R= as noted in table 1, age and gender did not differ significantly between the 12 cases and the 12 healthy controls. Weight at time of blood draw was not available for all cases and controls. All children were of Hispanic ethnicity. We do not collect information on ‘race’ in the underlying study as we do not feel that it would be meaningful or easily interpretable in the context of our population which is expected to have high and variable racial admixture. 

Further, in microarray experiments, variations of expression measurements can be attributed to many different sources that influence the stability and reproducibility of microarray platforms. How the authors handle this issue? 

R= One of the motivations to do this study is the remarkable stability of miRNAs in plasma. The samples have been collected under standard procedures and kept at 4°C until plasma separation (within 6 hours of sample collection) and then kept frozen at -70 °C until RNA isolation. Our analytic approach focused first on generating the detection score that simplifies handling microarray data, and later on examination of expression measurements. In our pilot work, we performed technical replicas, and since both expression and detection data were highly correlated, we opted to prioritize more biological replicas in our design instead of additional technical replicas. Expression measurements in microarray platforms are very sensitive and detect large amounts of ‘noise’. We believe that this sensitivity masks underlying data patterns which are unmasked by using the detection score. 

Is normalization applied when process the data?

R= Yes, we indicated this in the Data Analysis section Lines 147-148.

Results and discussions

There are several studies reported about 1000 to 2000 miRNAs in Rb patients’ plasma (https://www.ncbi.nlm.nih.gov/ pmc/articles/PMC5493862/). Why the authors only detected about 600 miRNAs in plasma?

R=to our knowledge this is the first work reporting the plasma miRNnome from children with retinoblastoma. The article mentioned by the reviewer is our own previous report of the primary tumors which correspond to the samples mentioned in this manuscript. We have cited this article in our references as reference ‘(20)’. We used the same platform as described in that publication: Affymetrix chips miRNA 4 with 2578 miRNAs.

We do not know why do we are able to detect approximately 600 miRNAs in plasma. We can only speculate the reasons. Plasma can be considered a ‘liquid’ tissue, and as such only a subset of genetic elements from the total miRNnome are “expressed” or detected. It may follow the same tenet as that of tissue specificity, which is result from the expression of only a subset of the genes in the genome. 

We realize that some information about the number of miRNAs per cluster shown in the detection map generated from the primary tumors is missing in Table 2. We have now added the corresponding data from the primary tumors. 

Rb tissues 995 1441 142

How the authors validate the miRNA expression profile? Any qRT-PCR validation applied?

R= we used stem-loops and exiqon miRNA probes for 6 miRNAs with high expression levels from the 19 miRNA signature for RB cases. We obtained mixed results and only 3 miRNAs showed high detection levels with both experimental approaches. We think that qRT-PCR is not optimal for miRNA detection and believe that more reliable methods need to be developed.

The figures’ resolution is always low, which are difficult to read.

R=we apologize for the difficulty in reading our images. With this revised submission, we submit better images with higher resolution. 

Reviewer #2: In this paper, the authors try to answer the questions what the differences are between the healthy person and Rb (diseased) person in terms of their miRNAs in plasma and extracellular vesicles using 12 patients and 12 controls. The size of miRNAs in plasma are 537 and in EV is 626. And they found that 19 miRNAs in plasma are significant between healthy and diseased person. And they also found 9 miRNAs that can be detected in all the samples, suggesting that these 9 miRNAs can be used as internal control. They also found that the gender is also a factor that affects the miRNAs between healthy and diseased person. 

First, I think the English in this paper needs to be improved. 

R= we apologize for the grammatical errors in our initial submission. We have carefully edited this revised submission and hope that it will be easier to read. 

Second, I think using 12 patients and 12 control as samples are too few to get conclusion. 

R= we recognize the sample size is small, however we believe our findings represent an important contribution that can advance the scientific field because we show a different analytic approach that opens up new possibilities for interpretation and permits extracting information that may potentially be useful in the field of using miRNAs as circulating biomarkers. We therefore consider our conclusions are informative and valid.

 Third, the difference between the plasmid and EV in terms of its chemical components should be discussed in the paper. 

R=the reviewer may refer to plasma instead of ´plasmid‘, we have now included the following text in the discussion: 

“Our finding that miRNA contents in total plasma and EVs are distinct, may be particularly informative. One possible explanation may be that the purification of EVs may amplify signals from miRNAs that may not be detectable in miRNAs extracted from whole plasma. Importantly, the 19 miRNA signature that is able to discriminate Rb cases from healthy controls is found only in RNA from total plasma and not in EVs, this is relevant because the method to obtain RNA from plasma is simpler and less expensive than the method to obtain RNA from of EVs.”

Other than that, I have some comments as below:

1. Methods/Patients and samples: What are including criteria and what are excluding criteria?

R= we have now included the following text in the Methods, patients and samples section: 

“Inclusion criteria: children newly diagnosed with retinoblastoma at an age of less than or equal to 6 years, without a known genetic syndrome or prior diagnosis of cancer, who were diagnosed or were receiving treatment at the two participating treatment centers, and did not have a preexisting/ known family history of retinoblastoma. One child later found to have familial retinoblastoma was included in the study. 

Exclusion criteria: age greater than 6 years of age at time of diagnosis, known diagnosis of a genetic syndrome or prior diagnosis of cancer.” 

2. How are the healthy control being selected?

R= Control selection used ‘friend’ controls as a proxy for population controls. Controls were children of women who were friends (but not a biological relative) of the mother of an eligible case child. Exclusion criteria for controls included being diagnosed with a known genetic syndrome, or having a family history of retinoblastoma, or a personal history of any type of cancer. Controls were within 2 years of age of the case child. We have included an abbreviated description of the source of the population at the beginning of the Materials and Methods section as we acknowledge that this should be included. 

3. How many times the experiments for Fig 1, 2, and 3 are done? If they are only done once, how do the authors make sure the results are reliable?

R= in our pilot work, we performed a technical replica for 1 tumor (2 microchips) and for 1 plasma (2 microchips), and since expression and detection data had a very high correlation, we decided to include in the design more biological replicas to obtain more information on reproducibility and biological variability. We performed tumor experiments 12 times, plasma experiments 24 times and EVs experiments 24 times, we believe these make our results reliable.

4. In figure 3 and 4, how top is the top ranked miRNAs? 

R=Figure 3 is a detection map generated with hierarchical clustering using the detection score for miRNAs isolated from EVs, there are no top ranked miRNAs in Fig 3.

Figure 4 shows heatmaps and volcano plots generated with the results of differential analysis using expression data. We used the expression ’top ranked’ to identify in the output results table, all significant over expressed miRNAs in the Rb cases. The results of Differential Analysis are lists of miRNAs with log2 transformed values, this transformation is performed to obtain symmetrical data, positive numbers for Rb cases, and negative numbers for healthy controls. We ordered this transformed values from large to small to identify at the top of positive values, those miRNAs over expressed in Rb cases.

To avoid confusion, we added the following:

In the differential analysis we found 158 differentially expressed miRNAs in plasma and 142 differentially expressed miRNAs in EVs. However, adjusted p values indicate that differences were not significant for any of them.

We also change the expression top ranked to ‘The most differentially expressed’ miRNAs in the table 3 and 4 titles. 

We also added before description of figure 4 the following:

The corresponding heatmap does not show color differences among cases and controls except for the 158 and 142 differentially expressed miRNAs which are shown in Fig 4. Even though differential analysis identified miRNAs with p-values <0.05, the adjusted p-values did not show any statistically significant differences between cases and controls for any miRNA. 

Are the top miRNAs showed in Table 3 and Table 4 same as those detected by mirnomic analysis?

R= Yes, the top miRNAs showed in Table 3 and Table 4 are the same as those indicated in the volcano plots in figure 4 b and d. We are submitting a better quality figure with larger fonts to enhance the readability of the miRNA identifiers.

5. In Figure 5 and table 5, the authors summarized the number of detected miRNAs per sample, per group (cases or controls) and per class. How does these numbers compare to other mirnome studies?

R=to our knowledge this is the first work reporting the size of plasma miRNome using the terms ‘detected’ miRNAs. We added in table 2 miRNome data from the corresponding primary tumors:

Rb tissues 995 1441 142

6. Figure 7 indicated that gender may be a factor that cause the miRNA to be different. So Could the authors analyze the difference of miRNA in plasma and EV between diseased person and health person with gender differences? So there will be four groups: 1) health female, 2) health male, 3) diseased female, and 4) diseased male.

R= We think that in order to interpret this findings, more balanced groups are needed with equal number of males and females in each group. Since our sex related findings were unexpected we didn’t consider this balance in the study design. Nonetheless we now submit the requested analysis and groups obtained with hierarchical clustering of detection scores and a summary of this analysis in the following table:

Groups Condition # miRNAs

shared

Group 1 Healthy females in Plasma 21

 Healthy females in EVs 33

Group 2 Healthy males in Plasma 22

 Healthy males in EVs 62

Group 3 Rb females in Plasma 63

 Rb females in EVs 73

Group 4 Rb males in Plasma 36

 Rb males in EVs 44

7. The authors found that 19 miRNAs shared exclusively by all Rb cases. Are these 19 miRNAs have any similarities? Which pathways did these 19 miRNAs targeted? How about this result compared with other similar results? Is there any difference? If yes, why?

R= from the 19 miRNAs in plasma that were shared by all Rb cases, 14 are also part of the 142 miRNA core shared by all the primary tumors described in (20). For the 19 miRNAs found, using mirPath v.3 we found 51 KEGG pathways with a wide variety of biologic functions which probably reflect the physiologic processes in which the circulating miRNAs are involved. There are no similar results in the literature.

8. Why “miR-16-5p signals are detectable in 17 / 21 samples mainly in EVs” but not all the samples in EVs? And why “miR-21 is detected only in a few samples” but not all the samples? Since miR-16 and miR-21 are all normalizers, they should be seen in all the samples?

R= miR-16 and miR-21 are the most common miRNA used as normalizers for plasma with qRT-PCR. Ideally a normalizer should be detectable in all the samples. Our data show that these two common normalizers are not detected in all samples. However, we found 9 miRNAs that were detectable in all the samples, making these nine better candidates to be used as normalizers. We do not know why miR-16 and miR-21 are not detectable in all of our samples.

9. Line 319: “13 miRNAs as a signature are exclusively detected in EVs from Rb cases.” Please delete exclusively because it is not exclusive as 12miRNAs are also seen in healthy controls.

R= we agree that the word produces confusion, our apology for this we eliminated it.

10. Why “more robust circulating Rb case signature is in the plasma compartment not in EVs”? Could the authors give an explanation?

R= this work essentially describes what can be detected in plasma and EVs using the detection score. At this moment we can only speculate about why the plasma signature clearly better discriminates Rb cases from healthy controls compared with the EVs signature. One possibility is that detectable miRNAs in plasma might reflect both the circulating tumor message as well as a systemic response to cancer.

11. Line 302-303: “the differences are found not too strong” is too subjective. Please revise this sentence, e.g., adding a reference.

R= We change the sentence as follows: 

“The corresponding heatmap does not show color differences among cases and controls except for the 158 and 142 differentially expressed miRNAs which are shown in Fig 4. Even though differential analysis identified miRNAs with p-values <0.05, the adjusted p-values did not show any statistically significant differences between Rb cases and controls for any miRNA.”

12. Table 1: Did the authors do the statistical analysis on the patient gender between the health and diseased group?

R= we compared the sex distribution between cases and controls and found no statistical difference. We have now added this in the footnote to the table.

13. EVs includes exosomes and microvesicles. In the introduction, the authors only introduce the exosomes but not the microvesicles. Please add some information of microvesicles.

 R= we arranged information in the introduction stating that: 

“In plasma, exosomes are part of extracellular vesicles (EVs) that include particles of diverse sizes and origins which are not fully characterized. There is not a standard definition of what exosomes are. In practical terms these are mixed populations of membranous nanosized particles about 50-130 nm in diameter (8)(9)(10).

---

## [Decision Letter · Decision Letter 1]

11 Mar 2020

PONE-D-19-34857R1

Plasma miRNome analysis from children with retinoblastoma reveals the size of the circulating miRNome, a tumor signature and novel candidates for normalizers.

PLOS ONE

Dear Dr. Castañeda,

Thank you for submitting your manuscript to PLOS ONE. After careful consideration, we feel that it has merit but does not fully meet PLOS ONE’s publication criteria as it currently stands. Therefore, we invite you to submit a revised version of the manuscript that addresses the points raised during the review process.

We would appreciate receiving your revised manuscript by Apr 25 2020 11:59PM. To enhance the reproducibility of your results, we recommend that if applicable you deposit your laboratory protocols in protocols.io, where a protocol can be assigned its own identifier (DOI) such that it can be cited independently in the future. For instructions see: http://journals.plos.org/plosone/s/submission-guidelines#loc-laboratory-protocols

We look forward to receiving your revised manuscript.

Kind regards,

Zhijing Tan, Ph.D

Academic Editor

PLOS ONE

Additional Editor Comments (if provided):

Castro-Magdonel et al., improved the manuscript following the reviewers' suggestion and respond to all comments one by one. There are two common issues in this manuscript. One is all tables should be three-line table. The second issue is all references should be formatted following Submission Guidelines, https://journals.plos.org/plosone/s/submission-guidelines#loc-references. All references should be double manually checked after formatting using common reference management software such as Endnote and disconnecting from the reference management software. For example, Ref. 3, Bartel DP. MicroRNAs. Cell [Internet]. 2004 Jan 23 [cited 2018 Apr 30];116(2):281–97. Available from:http://linkinghub.elsevier.com/retrieve/pii/S0092867404000455, could be formatted as: Bartel DP. MicroRNAs. Cell. 2004 Jan 23;116(2):281–97. In addition, subtitle Bibliography should be replaced by References.

Reviewers' comments:

Reviewer's Responses to Questions

**Comments to the Author**

1. If the authors have adequately addressed your comments raised in a previous round of review and you feel that this manuscript is now acceptable for publication, you may indicate that here to bypass the “Comments to the Author” section, enter your conflict of interest statement in the “Confidential to Editor” section, and submit your "Accept" recommendation.

Reviewer #1: All comments have been addressed

Reviewer #2: All comments have been addressed

2. Is the manuscript technically sound, and do the data support the conclusions?

Reviewer #1: Yes

Reviewer #2: Partly

3. Has the statistical analysis been performed appropriately and rigorously? 

Reviewer #1: Yes

Reviewer #2: Yes

4. Have the authors made all data underlying the findings in their manuscript fully available?

Reviewer #1: Yes

Reviewer #2: Yes

5. Is the manuscript presented in an intelligible fashion and written in standard English?

Reviewer #1: Yes

Reviewer #2: Yes

6. Review Comments to the Author

Reviewer #1: The authors have all my concerns addressed. There is no more questions on this manuscript. I suggest to accepting it for publication on PLOS One.

Reviewer #2: Thank you for taking care of review’s question. I have some comments:

1. Review 1/Question 1: the title still needs to be changed. Please specify how does the size of the circulating miRNome change and how does tumor signature change in the title.

2. Review 1/M&M: Please include “All children were of Hispanic ethnicity.” in the text.

3. Review 2/Question 1: please add explanation why year 6 is a cut-off line.

4. Review 2/Question 10: please add explanation in the text.

5. Review 3/Question 7 and 8: is the inconsistency with other literature and within the experiment an indication of something went not well in the experiment? For example, reproducibility?

7. PLOS authors have the option to publish the peer review history of their article (what does this mean?). If published, this will include your full peer review and any attached files.

Reviewer #1: Yes: Guobin Xia

Reviewer #2: No

---

## [Author Response · Author response to Decision Letter 1]

19 Mar 2020

Additional Editor Comments:

Castro-Magdonel et al., improved the manuscript following the reviewers' suggestion and respond to all comments one by one. There are two common issues in this manuscript. One is all tables should be three-line table. 

R= Following the Table guidelines we formatted the tables as requested and divided table 6 and thus we present now a table 7 and change the text accordingly.

The second issue is all references should be formatted following Submission Guidelines, https://journals.plos.org/plosone/s/submission-guidelines#loc-references. All references should be double manually checked after formatting using common reference management software such as Endnote and disconnecting from the reference management software. For example, Ref. 3, Bartel DP. MicroRNAs. Cell [Internet]. 2004 Jan 23 [cited 2018 Apr 30];116(2):281–97. Available from:http://linkinghub.elsevier.com/retrieve/pii/S0092867404000455, could be formatted as: Bartel DP. MicroRNAs. Cell. 2004 Jan 23;116(2):281–97. In addition, subtitle Bibliography should be replaced by References.

R= We double manually checked all references following Plos guidelines for references.

Reviewers' comments:

Reviewer's Responses to Questions

Comments to the Author

1. If the authors have adequately addressed your comments raised in a previous round of review and you feel that this manuscript is now acceptable for publication, you may indicate that here to bypass the “Comments to the Author” section, enter your conflict of interest statement in the “Confidential to Editor” section, and submit your "Accept" recommendation.

Reviewer #1: All comments have been addressed

Reviewer #2: All comments have been addressed

2. Is the manuscript technically sound, and do the data support the conclusions?

Reviewer #1: Yes

Reviewer #2: Partly

3. Has the statistical analysis been performed appropriately and rigorously? 

Reviewer #1: Yes

Reviewer #2: Yes

4. Have the authors made all data underlying the findings in their manuscript fully available?

Reviewer #1: Yes

Reviewer #2: Yes

5. Is the manuscript presented in an intelligible fashion and written in standard English?

Reviewer #1: Yes

Reviewer #2: Yes

6. Review Comments to the Author

Reviewer #1: The authors have all my concerns addressed. There is no more questions on this manuscript. I suggest to accepting it for publication on PLOS One.

Reviewer #2: Thank you for taking care of review’s question. I have some comments: 

1. Review 1/Question 1: the title still needs to be changed. Please specify how does the size of the circulating miRNome change and how does tumor signature change in the title.

Response= We recognize the requested modification of the title may better reflect the main contribution of this work to the field, we changed the title accordingly:

Circulating miRNome detection analysis reveals 537 miRNAS in plasma, 625 in extracellular vesicles and a discriminant plasma signature of 19 miRNAs in children with retinoblastoma from which 14 are also detected in corresponding primary tumors

2. Review 1/M&M: Please include “All children were of Hispanic ethnicity.” in the text.

R= we included all children were of Hispanic ethnicity in the “Patients and samples” section. We do not understand what M&M means.

3. Review 2/Question 1: please add explanation why year 6 is a cut-off line.

R= this is because the majority of newly diagnosed Rb cases fall under 6 year of age, we added the following in “Patients and samples” section: ...children newly diagnosed with retinoblastoma at an age of less than or equal to 6 years since most affected patients are in this age group.

4. Review 2/Question 10: please add explanation in the text.

R= We included the following in the “Discusion” section: At this moment we can only speculate about why the plasma signature clearly better discriminates Rb cases from healthy controls compared with the EVs signature. One possibility is that detectable miRNAs in plasma might reflect both the circulating tumor message as well as a systemic response to cancer,...

5. Review 3/Question 7 and 8: is the inconsistency with other literature and within the experiment an indication of something went not well in the experiment? 

R= Question 7 is related to the 19 miRNAs shared by all Rb cases. We previously explained that 51 KEGG pathways are affected by the 19 miRNA signature covering a wide biological group of functions. The fact that there are no other reports in the literature similar to our findings is not an inconsistency; it rather reflects the novelty of our analytical approach and findings. It is not clear for us what inconsistency within the experiment the reviewer refers to.

For example, reproducibility?

R= One of the motivations to do this study is the remarkable stability reported for miRNAs in plasma. In order to assure reproducibility the samples have been collected under standard procedures, kept at 4°C until plasma separation (within 6 hours of sample collection), and then kept frozen at -70 °C until RNA isolation. Our analytic approach focused first on generating the detection score that simplifies microarray data, and later on examination of expression measurements. In our pilot work, we performed technical replicas, and since both expression and detection data were highly correlated, we opted to prioritize more biological replicas in our design instead of additional technical replicas in order to find reproducible patterns shared by Rb cases. Expression measurements in microarray platforms are very sensitive and detect large amounts of ‘noise’. We believe that this sensitivity masks underlying data patterns, which are unmasked by using the detection score. 

R= we included the following in the “Discussion” section:

One of the motivations to do this study is the remarkable stability reported for miRNAs in plasma. In order to assure reproducibility the samples have been collected and processed under standard procedures. Our analytic approach focused first on generating the detection score that simplifies microarray data, and later on examination of expression measurements. In our pilot work, we performed technical replicas, and since both expression and detection data were highly correlated, we opted to prioritize and include more biological replicas in our design instead of additional technical replicas, to increase the possibilities of finding patterns shared by Rb cases. Expression measurements in microarray platforms are very sensitive and detect large amounts of ‘noise’, we believe that this sensitivity masks underlying data patterns, which are unmasked by using the detection score. 

Question 8 is related to miR-16-5p and miR-21, common normalizers used for qRT-PCR in plasma. We rather think our results are consistent with the literature adding precision to what is known, since we show that the compartment on which miR-16-5p is detected in most samples are EVs. We expand current knowledge by showing average detection levels as well as variability measurements (variance) across the samples under study. The contrast of this features particularly the expression levels between these common normalizers and the three most highly expressed newly proposed normalizers (miR-5787, 638 and 6732-5p) show that the later have higher detection levels and justify to continue the experimental exploration for using them as normalizers.

R= we included the following in the “Discussion” section: 

Our results add precision and expand information on miR-16-5p and miR-21 common normalizers for qRT-PCR in plasma, since we show that the blood compartment on which miR-16-5p is detected in most samples are EVs. We also show and contrast average detection levels as well as variability measurements (variance) across the samples under study. The expression levels between these common normalizers and the three most highly expressed newly proposed normalizers (miR-5787, 638 and 6732-5p) show that the later have higher detection levels and justify to continue the experimental exploration for using them as normalizers. 

7. PLOS authors have the option to publish the peer review history of their article (what does this mean?). If published, this will include your full peer review and any attached files.

Do you want your identity to be public for this peer review? For information about this choice, including consent withdrawal, please see our Privacy Policy.

Reviewer #1: Yes: Guobin Xia

Reviewer #2: No

---

## [Editor Report · Decision Letter 2]

24 Mar 2020

Circulating miRNome detection analysis reveals 537 miRNAS in plasma, 625 in extracellular vesicles and a discriminant plasma signature of 19 miRNAs in children with retinoblastoma from which 14 are also detected in corresponding primary tumors

PONE-D-19-34857R2

Dear Dr. Castañeda,

We are pleased to inform you that your manuscript has been judged scientifically suitable for publication and will be formally accepted for publication once it complies with all outstanding technical requirements.

With kind regards,

Zhijing Tan, Ph.D

Academic Editor

PLOS ONE

Additional Editor Comments (optional):

None
---

## [Editor Report · Acceptance letter]

31 Mar 2020

PONE-D-19-34857R2 

Circulating miRNome detection analysis reveals 537 miRNAS in plasma, 625 in extracellular vesicles and a discriminant plasma signature of 19 miRNAs in children with retinoblastoma from which 14 are also detected in corresponding primary tumors 

Dear Dr. Ponce-Castañeda:

I am pleased to inform you that your manuscript has been deemed suitable for publication in PLOS ONE. Congratulations! Your manuscript is now with our production department. 

With kind regards,

on behalf of

Dr. Zhijing Tan 

Academic Editor

PLOS ONE